# Associative learning changes cross-modal representations in the gustatory cortex

Roberto Vincis*, Alfredo Fontanini*

Department of Neurobiology and Behavior, State University of New York at Stony Brook, Stony Brook, United States

**Abstract** A growing body of literature has demonstrated that primary sensory cortices are not exclusively unimodal, but can respond to stimuli of different sensory modalities. However, several questions concerning the neural representation of cross-modal stimuli remain open. Indeed, it is poorly understood if cross-modal stimuli evoke unique or overlapping representations in a primary sensory cortex and whether learning can modulate these representations. Here we recorded single unit responses to auditory, visual, somatosensory, and olfactory stimuli in the gustatory cortex (GC) of alert rats before and after associative learning. We found that, in untrained rats, the majority of GC neurons were modulated by a single modality. Upon learning, both prevalence of cross-modal responsive neurons and their breadth of tuning increased, leading to a greater overlap of representations. Altogether, our results show that the gustatory cortex represents cross-modal stimuli according to their sensory identity, and that learning changes the overlap of cross-modal representations.

*For correspondence: roberto. vincis@stonybrook.edu (RV); alfredo.fontanini@stonybrook.edu (AF)

**Competing interests:** The authors declare that no competing interests exist.

## Introduction

Traditionally, sensory cortices have been studied for their ability to encode stimuli of a single sensory modality. However, a growing body of literature has shown that activity in primary sensory cortices can also be modulated by cross-modal stimuli (*Ghazanfar and Schroeder, 2006*). Neurons in the visual cortex respond to acoustic stimuli (*Iurilli et al., 2012*) and activity in the auditory cortex is also shaped by visual (*Calvert et al., 1997*; *Brosch et al., 2005*) and somatosensory stimulation (*Zhou and Fuster, 2000*; *Schroeder et al., 2001*; *Fu et al., 2003*; *Brosch et al., 2005*; *Lakatos et al., 2007*). Similar results have been observed in the olfactory (*Wesson and Wilson, 2010*; *Maier et al., 2012*, *2015*), gustatory (*De Araujo and Rolls, 2004*; *Samuelsen et al., 2012*; *Gardner and Fontanini, 2014*) and somatosensory cortices (*Zhou and Fuster, 2000*). While these studies have been fundamental in demonstrating cross-modality and its potential functions (*Ghazanfar and Schroeder, 2006*; *Samuelsen et al., 2012*; *Gardner and Fontanini, 2014*; *Stein et al., 2014*; *Kusumoto-Yoshida et al., 2015*; *Yau et al., 2015*), several key issues regarding the representation of cross-modal stimuli remain largely unaddressed. Specifically, there are two pressing questions that we investigated in this study.

The first question pertains to the neural representation in a primary sensory cortex of multiple stimuli belonging to different sensory modalities. Studies from the literature relied mostly on cross-modal stimuli of a single or a couple of modalities (*Ghazanfar et al., 2005*; *Kayser and Logothetis, 2007*; *Lakatos et al., 2007*; *Driver and Noesselt, 2008*; *Vasconcelos et al., 2011*; *Iurilli et al., 2012*). As a result, it is not known whether cross-modal stimuli pertaining to distinct sensory modalities converge on the same set of sensory cortical neurons or whether they recruit different groups of neurons. The former possibility is compatible with cross-modal responses reflecting a general, modality-independent arousal or distracting signal (*Talsma et al., 2010*; *Iurilli et al., 2012*) and suggests the existence of broadly tuned, multi-modal neurons capable of being activated by any cross-

**eLife digest** Imagine that you are waiting for a cappuccino at your favorite café. You hear the sound of the steamer, and shortly afterwards the barista calls your name and announces that your cappuccino is ready. As they hand it to you, you see the foam sprinkled with cocoa and the aroma of the cappuccino reaches your nose. You can almost taste it. When you finally take your first sip, the taste is hardly a surprise; it is just as your eyes and nose predicted.

How does the brain deal with such a rich and multisensory experience? How does it learn to associate the sight and smell of a cappuccino with its taste? Specialized regions of the brain called associative areas were traditionally thought to perform this task. These areas receive inputs from every sensory system and can link information from these different sources together. According to this view, the job of each individual sensory system is to pass along information relevant to one particular sense.

More recent results, however, challenge this strict division of labor and suggest that individual sensory systems may be able to combine information from multiple senses. Thus the sights, sounds and odors associated with our cappuccino may also activate the area of the brain in charge of processing taste: the gustatory cortex. To investigate this possibility, Vincis and Fontanini set out to determine whether neurons in the gustatory cortex of rats can process stimuli belonging to senses other than taste.

As predicted, neurons in the gustatory cortex did change their firing rates in response to odors, touch, sounds and light. However, more of the gustatory neurons responded to odors and touch than to sounds and light. In addition, of the four stimuli, the rats most easily learned to associate odors and touch with a sugary solution. This is consistent with the fact that rodents rely more upon their whiskers and their sense of smell to find food they do their eyes and ears. Finally, learning to associate a stimulus other than taste with a sugary solution increased the number of neurons in the gustatory cortex that subsequently responded to other senses and changed their response properties.

Further studies are now required to answer three questions. Why can some senses more effectively influence the activity of the gustatory cortex than others? Can gustatory neurons distinguish between different stimuli of the same type – different odors, for example? What are the neural pathways that convey multisensory information to the gustatory cortex? Answering these questions will help us to better understand how sensory systems link information from multiple senses.

modal stimulus. The latter possibility proposes that primary sensory areas can represent the sensory modality of cross-modal stimuli and predicts the presence of stimulus-specific unimodal representations.

The second question focuses on whether cross-modal representations can be shaped by associative learning. Most of the studies in the literature relied on cross-modal stimuli that had not been explicitly associated with the modality of the area under investigation (*Ghazanfar and Schroeder, 2006*; *Kayser et al., 2007*). Previous studies focusing on sensory neocortices and on the gustatory cortex suggested that associative learning could dramatically change how predictive cues modulate cortical neurons (*Zhou and Fuster, 2000*; *Brosch et al., 2005*; *Samuelsen et al., 2012*; *Gardner and Fontanini, 2014*). These studies, however, relied on predictive cues pertaining to a few sensory modalities. As a result, little information is available on the effects of associative learning on the neural representations of cross-modal stimuli of all sensory modalities.

In this study, we addressed these two fundamental questions by studying cross-modal responses in the gustatory cortex (GC) – an area that represents a good model for studying cross-modality (*Katz et al., 2002*; *Simon et al., 2006*; *Carleton et al., 2010*; *Veldhuizen et al., 2010*). We presented stimuli belonging to four sensory modalities: auditory, visual, somatosensory, or olfactory. Cross-modal stimuli were presented to two, distinct groups of animals: a group not trained to associate cross-modal stimuli with sucrose (or any other taste) and another in which rats were trained to associate cross-modal stimuli with sucrose. The first cohort of rats served to determine if GC could

respond to the cross-modal stimuli used and to characterize their baseline representations. The second cohort allowed us to determine if and how associative learning modified baseline cross-modal representations. Single unit recordings in alert, untrained rats revealed that GC neurons could represent the identity of cross-modal stimuli via largely non-overlapping representations and show a strong bias toward olfactory and somatosensory stimuli. Olfactory and somatosensory stimuli also led to faster associative learning compared to visual and auditory stimuli. Comparisons between neural responses in untrained and trained rats demonstrated that, upon learning, GC maintained the ability to discriminate different sensory modalities, while also developing associative responses. Association with a common gustatory outcome increased the prevalence of neurons responding to cues, widened the overlap between neural representations and enhanced the similarity between responses to cues with similar associability.

## Results

### Representation of cross-modal stimuli in GC

We first investigated how GC neurons encode cross-modal stimuli that have not been explicitly associated with taste. We recorded single-neuron spiking activity and orofacial movements of alert rats (*Figure 1—figure supplement 1A*; n = 5; untrained) in response to random deliveries of the following sensory stimuli: an odor, a puff of air on the distal portion of the rats' whiskers (*air puff*), a light and a tone. Sucrose, NaCl, citric acid and quinine were also delivered to test each neuron's response to taste. (*Figure 1—figure supplement 1*; see Materials and methods). Gustatory stimuli were delivered intraorally via a manifold inserted into an intra-oral cannula. Intraoral delivery was chosen for complete control of stimulus delivery and for consistency with previous studies on expectation in GC (*Samuelsen et al., 2012*, *2013*; *Gardner and Fontanini, 2014*; *Liu and Fontanini, 2015*). The long and variable inter trial interval (*Figure 1—figure supplement 1B*; 50 ± 10 s) was designed to avoid associations between cross-modal, 'non-gustatory' stimuli and tastants.

To assess single neuron responses to cross-modal and gustatory stimuli, we analyzed stimulus-evoked changes in firing rates using a change point analysis (*Jezzini et al., 2013*; *Liu and Fontanini, 2015*; *Mazzucato et al., 2015*). Briefly, for each neuron, we computed the cumulative distribution function (CDF) of spike occurrences across all trials within a temporal window starting 1 s before stimulus onset (i.e., baseline) and ending 2 s (for cross modal stimuli) or 2.5 s (for gustatory stimuli) after stimulus presentation. Sudden changes in firing rates resulted in a change in the CDF slope and in the identification of a 'change point' (CP). A binomial test (p<0.05) was used to compare spike counts before and after each CP to evaluate their statistical significance (*Gallistel et al., 2004*). This method was used because of its sensitivity and reliability.

As expected, we observed that a substantial percentage of neurons significantly changed their firing rates following gustatory stimulation (*Figure 1A*; 38.5% [52/135], 41.4% [56/135], 41.4% [56/135] and 38.5% [52/135] for sucrose, NaCl, citric acid and quinine, respectively). In addition, a consistent proportion of neurons were modulated by other sensory modalities (*Figure 1A*; 33.3% [45/135] of recorded neurons responded to at least one non-gustatory stimulus; odor: 16.2% [22/135]; air puff: 15.5% [21/135]; tone: 4.4% [6/135]; light: 3.7% [5/135]). Orofacial movements evoked by non-taste stimuli were hardly detectable and significantly smaller compared to the ones evoked by tastants, indicating that they were not driving the changes in neuronal activity we observed (*Figure 1—figure supplement 1C*; Wilcoxon sign rank test: p<0.05). The absence of a relationship between responses to cross-modal stimuli and mouth movements can also be visualized for the two representative neurons depicted in *Figure 1D–G*. Among the non-gustatory stimuli, odors and somatosensory stimuli appeared to be the most effective in modulating GC neurons. When comparing the prevalence of cross-modal responses for the different stimuli, we observed significant differences in the proportion of neurons that responded to cross-modal stimuli (Pearson's $\chi^2$ test; $\chi^2_{(3)}$ = 17.35, p<0.001). Specifically, while the proportion of neurons responding to odorant and air puff were similar (*Figure 1A*; Marascuillo's test, p>0.05), both were significantly higher compared to the proportion of neurons responding to sound and light (*Figure 1A*; Marascuillo's test, p>0.05). We then investigated the convergence of stimuli of different sensory modalities onto single GC neurons. We first identified, within the group of neurons whose firing rates were modulated by at least one taste delivery, those that were taste selective. A neuron was deemed to be taste selective if it showed significantly different

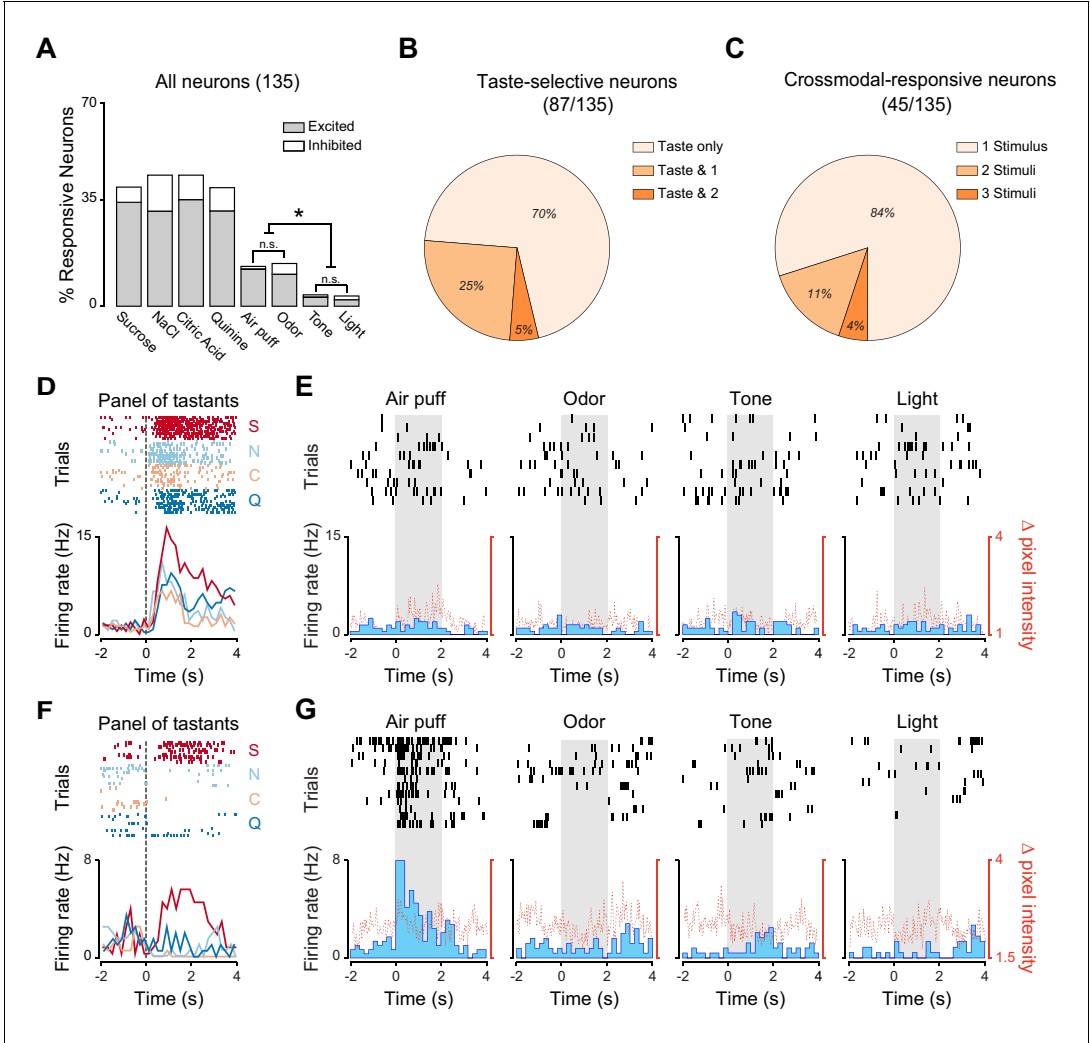

**Figure 1.** Neural representation of different sensory modalities in GC of untrained rats. (**A**) Percentage of neurons (n = 135) modulated (excited [gray] or inhibited [white]) by each stimulus. Odor and somatosensory (air puff) stimuli modulate a larger number of neurons compared to tone and light. Asterisks indicate *post-hoc* test corrected for multiple comparisons (Marascuillo's test, p<0.05). (**B**) Pie chart showing the proportion of taste selective neurons (n = 87), that are modulated exclusively by taste (Taste only) or by taste and either one (Taste & 1) or two (Taste & 2) cross-modal stimuli. No neuron was modulated by taste and three or four cross-modal stimuli. (**C**) Pie chart showing the proportion of cross-modal responsive neuron (n = 45), modulated by one (1 Stimulus), two (2 Stimuli) or three (3 Stimuli) out of the four stimuli used. No neuron was modulated by all four the stimuli. (**D**) Raster plots and PSTHs of a representative GC neuron showing a broadly tuned response to the four tastants. (**E**) Raster plots and PSTHs of a GC neuron (same as the one showed in panel D) showing no response to cross-modal stimuli (air puff, odor, tone and light). (**F**) Raster plots and PSTHs of a second representative GC neuron showing a selective response to taste. (**G**) Raster plots and PSTHs of a GC neuron (same as the one showed in panel D) showing a selective response to a puff of air and no responses to Odor, Tone and Light. For D and F: vertical lines at time 0 indicate the onset of the stimulus. S for sucrose, N for NaCl, C for citric acid and Q for quinine. For E and G: grey shaded areas indicate the time and duration of cross-modal stimulus presentation. Red dotted line represents the time-course of orofacial movement activity.

The following source data and figure supplement are available for figure 1:

**Source data 1.** Percentage of GC neurons responding to cross-modal and gustatory stimuli in untrained rats.

**Figure supplement 1.** Electrodes placement and experimental design for the group of untrained rats.

responses to the four gustatory stimuli (*Jezzini et al., 2013*; *Samuelsen et al., 2013*). Within the group of taste selective neurons (87/135), the majority responded exclusively to gustatory stimuli (*Figure 1B, D–E*; 70.1%, [61/87]), with only a few also activated by one (*Figure 1B, F–G*; 25.3%, [22/87]) or two (*Figure 1B*; 4.6%, [4/87]) cross modal stimuli. The percentage of neurons being both taste selective and responsive to cross-modal stimuli was not significantly different from what expected if the probabilities of the two conditions were independent (bootstrap procedure, p>0.05). We then turned our attention to the group of neurons responsive to cross-modal stimuli (45/135) and analyzed the convergence of non-gustatory responses (*Figure 1C*). The majority of cross-modal neurons were activated exclusively by one of the non-taste stimuli (either air puff, odor, tone or light stimuli; *Figure 1G*; 84.4%, [38/45]) and only a few neurons responded to two (11.1%, [5/45]) or three (4.4%, [2/45]) cross-modal stimuli. These percentages were not significantly different from those expected by chance given the fractions of neurons responding to different cues (bootstrap procedure, p>0.05 for all cases).

Altogether, our data indicate that GC neurons are modulated by tastants and by stimuli of other sensory modalities. Among non-tastants, olfactory and somatosensory stimuli recruited more neurons compared to visual and auditory stimuli. The representations of stimuli belonging to different sensory modalities overlapped only minimally.

## Learning associations between cross-modal stimuli and sucrose

We investigated if the different cross-modal stimuli were equally effective in driving learning of cue-sucrose associations. A second group of rats (trained, n = 5) was conditioned to associate each of the four cross-modal, 'non-gustatory' sensory cues with the intraoral delivery of sucrose (the unconditioned stimulus). In every daily session (for a total of 14 days), rats experienced twenty trials of cue-sucrose pairings for each modality (*Figure 2—figure supplement 1A*; see Materials and methods). For each trial, the sensory modality of the cue presented was chosen in a pseudo-random manner. In order to track associative learning we monitored orofacial movements in response to conditioned stimuli (*Figure 2—figure supplement 1B*). A significant increase of mouth movements during the cue compared to baseline indicated the appearance of a conditioned response and allowed us to establish learning (*Figure 2A leftmost panels*). Analysis of orofacial movements revealed that after 14 days of training rats learned to associate each anticipatory cue with the delivery of sucrose (*Figure 2A*). The conditioned orofacial movements were significantly stronger on the last day of training compared to the first day (*Figure 2A*; somatosensory: two-way ANOVA [factors: days and temporal epochs], $F_{(1,2)} = 43.8$, p<0.01; odor: two-way ANOVA, $F_{(1,2)} = 45.3$, p<0.01; tone: two-way ANOVA, $F_{(1,2)} = 47.7$, p<0.01; light: two-way ANOVA, $F_{(1,2)} = 76.6$, p<0.01). *Post-hoc* analyses revealed that, for each sensory modality, the difference in mouth movements across days was specific to the cue presentation epoch (*Figure 2A*; 'cue': p<0.01 for days effect with Tukey's correction). No difference across days was observed for spontaneous and taste-evoked mouth movements (*Figure 2A*; 'spontaneous': somatosensory: p=0.4, odor: p=0.9, tone: p=0.9, light: p=0.9; 'taste': somatosensory: p=0.9, odor: p=0.9, tone: p=0.8, light: p=0.9; Tukey's correction).

We then studied how cue-taste learning unfolds for the different sensory modalities over 14 days. This analysis was performed to determine if cross-modal cues different in associability. Since odor and tactile stimuli were more represented in GC before any cue-taste association was established (*Figure 1*), we hypothesized that this bias may predict a difference in learning rate. Indeed, odor and somatosensory cues evoked mouth movements significantly larger than baseline (defined as orofacial movements on the first day of conditioning) on the third day of training (*Figure 2B*; p<0.05 with Tukey's correction). In contrast, visual and auditory cues required seven days of conditioning to display significant changes in orofacial movements (*Figure 2B*; p<0.05 with Tukey's correction). Regardless of the difference in learning rate, the magnitude of the mouth movements evoked by each anticipatory cue was not significantly different on the last day of conditioning (one-way ANOVA [factor: cross-modal cues], $F_{(3)} = 2.24$, p=0.12).

In conclusion, analysis of conditioned responses revealed that each cross-modal stimulus could be successfully associated with taste within 14 days. Odors and somatosensory stimuli have a higher associability to sucrose compared to visual and auditory cues.

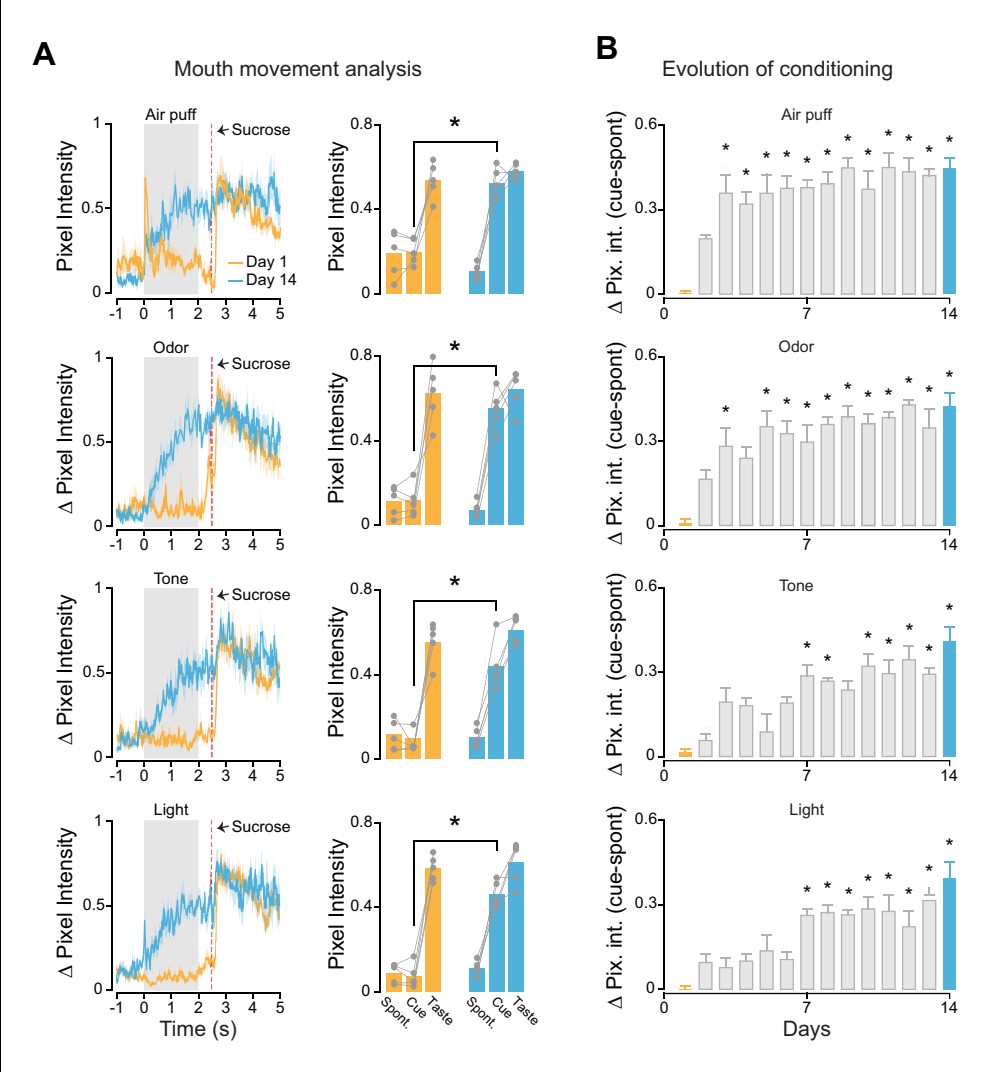

**Figure 2.** Associative learning for different cross-modal stimuli. (**A**) Left most panel: Average time course of orofacial movements evoked by cross-modal cues and sucrose during the first (gold) and 14th (cyan) day of classical conditioning (n = 5 rats). Shaded grey area and red dotted line indicate cues and sucrose presentation respectively. The shaded area around the curve represents ± SEM. Right most panel: average across trained rats (n = 5) of mouth movements (assessed by changes in pixel intensity) in different temporal epochs (1 s before cue onset [−1 to 0 s; Spont.], 1 s after cue onset [0.5–1.5 s; Cue] and 1 s after sucrose onset [2.7–3.7 s; Taste]) for the first (gold) and 14th (cyan) day of classical conditioning. (**B**) Average (n = 5 rats) of mouth movements across the 14 conditioning days. For A and B: asterisks indicate *post-hoc* tests corrected for multiple comparisons (Tukey, p<0.05). Rows from 1 to 4 show the orofacial activity evoked by air puff, odor, tone and light respectively.

The following source data and figure supplement are available for figure 2:

**Source data 1.** Values of the orofacial movements evoked by the cross-modal stimuli.

**Figure supplement 1.** Experimental design for cue-taste association experiments and assessment of conditioned responses.

## Representation of cues after associative learning

In order to establish if associative learning changes how GC represents information pertaining to cross-modal stimuli, neural activity was recorded after 14 days of conditioning (*Figure 3*). Random passive intraoral delivery of one out of four tastants during the experimental session allowed us to

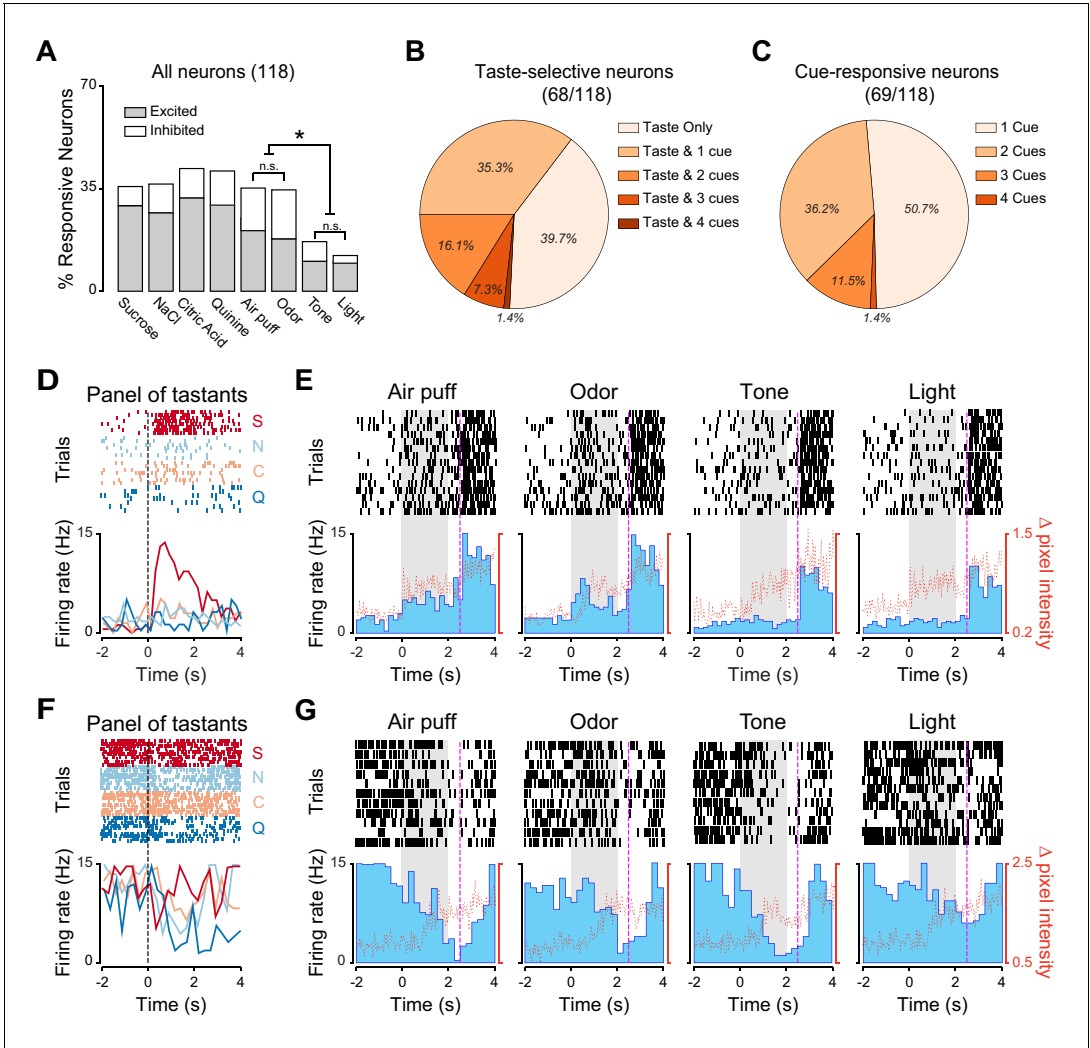

**Figure 3.** Neural representation of different sensory modalities after cue-taste association. (A) Percentage of neurons (n = 118) modulated (excited [gray] or inhibited [white]) by each tastants and anticipatory cues. Asterisks indicate *post-hoc* test corrected for multiple comparisons (Marascuillo's test, p<0.05). (B) Pie chart showing the proportion of taste selective neuron (n = 68) that are modulated exclusively by taste (Taste only) or by taste and one (Taste & 1), two (Taste & 2), three (Taste & 3) or four (Taste & 4) anticipatory cues. (C) Pie chart showing the proportion of cue responsive neuron (n = 69), which are only modulated by one (1 Cue), two (2 Cues), three (3 Cues) or all anticipatory cues (4 Cues). (D) Raster plot and PSTH of a representative GC neuron showing a significant excitatory response to sucrose. (E) Raster plot and PSTH of a GC neuron (same as the one showed in panel D) featuring significant excitatory responses to Air puff and Odor and no responses to Tone and Light. (F) Raster plot and PSTH of a GC neuron showing inhibitory responses to multiple tastants. (G) Raster plot and PSTH of a GC neuron (same as the one showed in panel F) displaying significant inhibitory responses to multiple cues (air puff, odor, tone and light). For D and F: vertical lines at time 0 indicate the onset of the taste delivery. S for sucrose, N for NaCl, C for citric acid and Q for quinine. For E and G: gray rectangular areas indicate the period of cue presentation. Vertical dotted lines at time 2.5 indicate the onset of sucrose delivery. Red dotted line represents the time-course of orofacial movement activity.

The following source data and figure supplements are available for figure 3:

**Source data 1.** Percentage of GC neurons responding to cross-modal and gustatory stimuli in trained rats.

**Figure supplement 1.** Electrode placement and experimental design for the group of trained rats.

**Figure supplement 2.** The neural bias for somatosensory and olfactory stimuli does not depend on the number of sessions at final performance level.

define if a GC neuron was taste selective (*Figure 3—figure supplement 1*). Comparison of responses to cross-modal stimuli between trained and untrained animals revealed significant differences. Indeed, cue-taste association correlated with a significantly higher prevalence of neurons that responded to cues of different sensory modalities compared to the group of untrained rats (*Figure 4A*; test for equality of proportion for all comparison; odor: 16.2% [22/135] untrained *vs.* 33.0% [39/118] trained, $\chi^2_{(1)} = 9.65$, p<0.01; air puff: 15.5% [21/135] untrained *vs.* 34.7% [41/118] trained, $\chi^2_{(1)} = 12.53$, p<0.01; tone: 4.4% [6/135] untrained *vs.* 16.1% [19/118] trained, $\chi^2_{(1)} = 9.60$, p<0.01; light: 3.7% [5/135] untrained *vs.* 11.8% [14/118] trained, $\chi^2_{(1)} = 6.03$, p<0.05). The increase was specific for responses to cross-modal stimuli, as learning did not change the prevalence of taste selective neurons (64.4% [87/135] for untrained *vs.* 57.6% [68/118] for trained, test for equality of proportion, $\chi^2_{(1)} = 1.23$, p=0.26). Similarly conserved in trained rats was the bias for olfactory and somatosensory stimuli; indeed, significantly more neurons responded to air puff and odor compared to light and sound (*Figure 3A*, Marascuillo's test, p>0.05).

Could the post-learning bias for olfactory and somatosensory stimuli depend on the fact that they had been learned faster than visual and auditory stimuli and hence had been presented for more sessions after the establishment of a cue-sucrose association? Additional analyses were performed to determine if the distribution of neurons responsive to cross-modal stimuli depended on the number of sessions spent at the final performance level. To address this issue we computed the fraction of neurons responsive to cross-modal stimuli for sessions recorded on the same number of days (i.e., 12, 13 and 14 days) after reaching final performance level (i.e., days 15, 16, 17 for odor and air puff and days 19, 20, 21 for tone and light). As *Figure 3—figure supplement 2* shows, the bias toward olfactory and somatosensory stimuli persisted within this matched sample of sessions. In addition, a second analysis was performed to determine directly if the number of cue responsive neurons increased over the course of subsequent sessions (*Figure 3—figure supplement 2*). No increase and no difference in the number of cross-modal responsive neurons were observed with increasing the number of sessions at final performance level. These results demonstrate that the magnitude of cross-modal representations is not related to exposition to the stimulus after learning.

Next, the convergence of stimuli belonging to different sensory modalities onto single GC neurons was analyzed in trained animals. First, we focused our attention on taste selective neurons (*Figure 3B*). Out of 68 neurons that were taste selective, 39.7% were activated exclusively by gustatory stimulation (*Figure 3B*; 27/68). The remaining taste selective neurons were also activated by one or more anticipatory cues; specifically 35.3% (24/68) were modulated by one cue, 16.1% (11/68) by two, 7.3% (5/68) by three and 1.4% (1/68) by all four. The proportion of neurons that were both taste selective and cue responsive is consistent with what expected from the independent probabilities of the two conditions (bootstrap procedure, p>0.05). However, and more importantly, the convergence between gustatory and cross-modal information was significantly different between

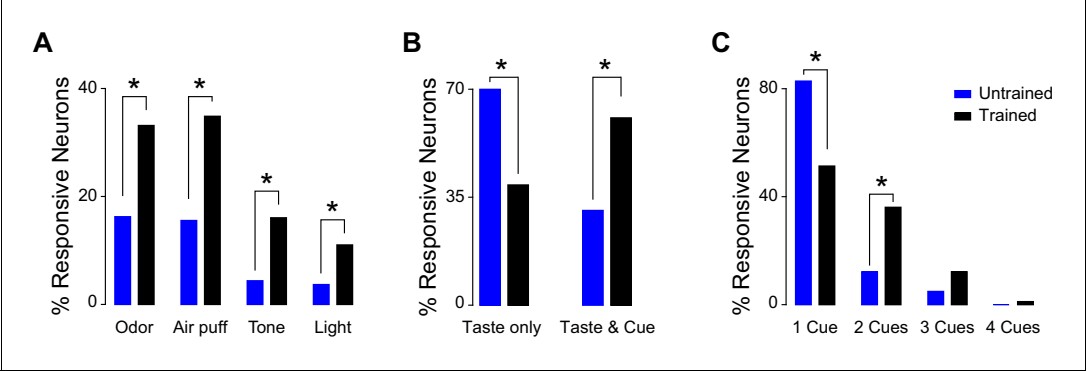

**Figure 4.** Comparison of cross-modal GC responsiveness between untrained and trained animals. (A) Percentage of GC neurons that are modulated by cross-modal stimuli in untrained (blue) and trained (black) rats. (B) Percentage of taste selective GC neurons that are modulated only by tastants (Taste Only) or by tastants and at least one cross-modal stimulus (Taste & Cue); blue: untrained, black: trained. (C) Percentage of cue responsive GC neurons that are modulated only by one or more anticipatory cross-modal cues in untrained (blue) and in trained (black) rats. Asterisk indicates p<0.05 for test for equality of proportion.

untrained and trained animals. When compared with untrained rats, trained rats showed a significant reduction in the number of single units responsive exclusively to taste (*Figure 4B*; 70.1% [61/87] for untrained and 39.7% [27/68] for trained rats: test for equality of proportion; $\chi^2_{(1)}$ = 14.38, p<0.001) and a significant increase in the number of neurons responding to both taste and cues (*Figure 4B*; 33.3% [29/87] for untrained and 60.2% [41/68] for trained rats: test for equality of proportion; $\chi^2_{(1)}$ = 11.20, p<0.001). The latter results are consistent with the increase in cue responsive neurons observed after associative learning.

We then investigated how many cue responsive neurons were activated by more than one cue. Out of 69 neurons, which responded at least to one cross-modal stimulus, half of them (50.7%, [35/69]) were activated only by one out of four anticipatory cues (*Figure 3C*). However, a substantial amount of cue responsive neurons were also responsive to two (36.2%, [25/69]), three (11.5%, [8/69]) and four (1.4%, [1/69]) cues. The number of neurons responding to more than one cue was not different from what predicted from the individual probabilities of occurrence (bootstrap procedure, p>0.05 for all cases). More relevantly, there was a significant difference in the percentage of neurons responding to one and two cues between the untrained and trained groups (*Figure 4C*). Specifically, in the trained groups of rats, we observed fewer neurons that were responsive only to one cue (*Figure 4C*; untrained: 84.4% [38/45]; trained: 50.7% [35/69]; test for equality of proportion; $\chi^2_{(1)}$ = 9.16, p<0.01), and more neurons that were activated by two cues (*Figure 4C*; untrained: 11.1% [5/45]; trained: 36.2% [25/69]; test for equality of proportion; $\chi^2_{(1)}$ = 5.60, p<0.05). *Figure 3D–G* show two examples of GC neurons that are taste selective and modulated by 2 (*Figure 3D–E*) and 3 (*Figure 3F–G*) cross-modal cues.

We performed an additional analysis in the group of trained rats to establish whether neurons that responded to cross-modal cues displayed a preference for any of the tastants. No bias toward any specific tastant was observed (cues/unexpected-sucrose 40.5% [28/69], cues/NaCl 37.6% [26/69], cues/citric acid 43.4% [30/69], cues/quinine 44.9% [31/69]; Pearson's $\chi^2$ test; $\chi^2_{(3)}$ = 0.36, p=0.94). This result suggests that in trained animals, cue responsive neurons do not preferentially represent sucrose, but maintain the ability to represent multiple tastants.

To establish whether the changes described above modified breadth of tuning, we computed an index of response sharpness (*Yoshida and Katz, 2011*). If a GC neuron is modulated only by one stimulus, its sharpness index is 1; if a neuron is responsive to all 8 stimuli, its sharpness index is 0. The results confirm that associative learning broadened the response tuning for both taste selective and cross-modal responsive neurons (taste selective neuron: 0.58 ± 0.02 and 0.48 ± 0.02 untrained and trained respectively, t-test: $t_{(152)}$ = −3.31; p<0.01; cross-modal responsive neurons: 0.60 ± 0.02 and 0.46 ± 0.01 untrained and trained respectively, t-test: $t_{(107)}$ = −4.77; p<0.01).

Overall, these results show that, when associated with sucrose, cues of different sensory modalities recruit more neurons in GC compared to untrained animals. Cross-modal cue-taste association does not eliminate the bias for stimuli with strong associability, but enhances the overlap between representations of multiple cues and between representations for cues and taste.

## Temporal analysis of cue responses

GC neurons are known to encode taste through dynamic changes in firing rates (*Katz et al., 2001*; *Jones et al., 2007*; *Jezzini et al., 2013*). We performed a series of analyses to investigate the time-course of cross-modal cue responses. *Figure 5A* shows the normalized firing rate (auROC) responses for all the neurons that were modulated by at least one anticipatory cue in trained and untrained rats. Visual inspection of these plots suggests that associative learning may be associated with an increase in the number of GC neurons that are inhibited by the cues, and may lead to more time-varying neural responses. We quantified the number of neurons that were either excited or inhibited (only the first change in firing compared to baseline is considered here) by the cue in untrained and trained rats. In the group of trained animals we observed a reduction in the proportion of neurons that were excited by the cue (*Figure 5A–B*; 70.3% [38/54] for untrained *vs.* 52.2% [59/113] for trained, test for equality of proportion; $\chi^2_{(1)}$ = 4.30, p<0.05) and an increase in the proportion of neurons that were inhibited by the cue (*Figure 5A–B*; rightmost panel; 29.6% [16/54] for untrained *vs.* 47.7% [54/113] for trained, test for equality of proportion; $\chi^2_{(1)}$ = 4.94, p<0.05). To determine whether learning was associated with responses with more frequent firing rate changes, we computed the average number of firing rate modulations for each response. In the group of trained rats, cues evoked a significantly higher number of firing rate modulation per response compared to what

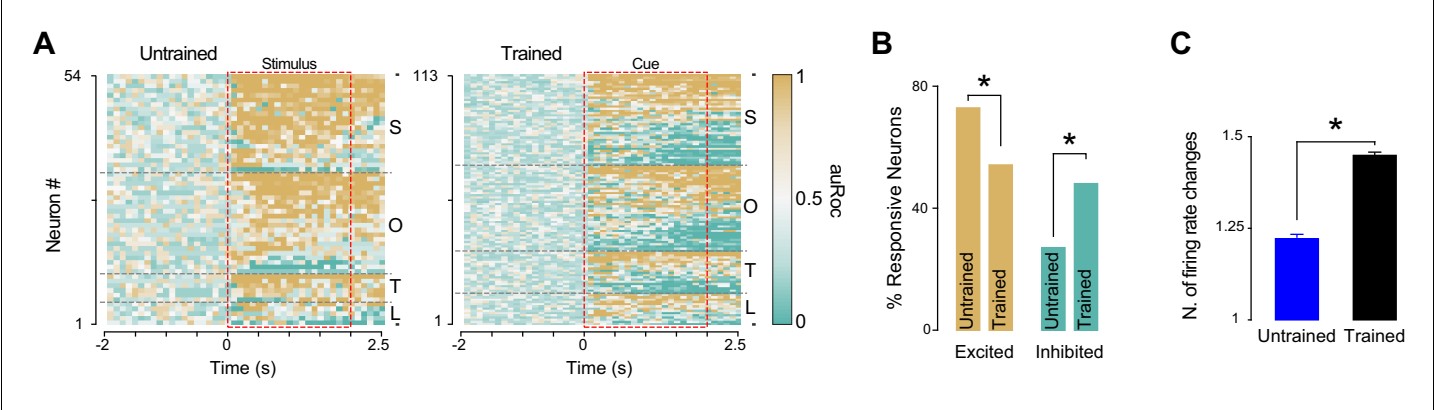

**Figure 5.** Time course of GC responses in the groups of trained and untrained rats. (**A**) Population plot of all GC neurons modulated by at least one cross-modal stimulus in untrained (left panel) and in trained (right panel) rats. Each row represents a GC neuron. The color of each square along the x-axes (see color bar) represents the normalized firing rate within each 200 ms bin. The red dotted rectangular box indicates stimulus presentation. Neurons are clustered by the sensory modality (S for somatosensory, O for odor, T for tone and L for light) and ranked with excitatory and inhibitory responses from top to bottom respectively. (**B**) Histograms showing the percentage of cue responsive neurons that are excited and inhibited by anticipatory cues. (**C**) Histogram showing the average number of firing rate modulations in neurons responsive to cross-modal stimuli. Asterisk indicates p<0.05 for test for equality of proportion and t-test for panel B and C respectively.

The following source data is available for figure 5:

**Source data 1.** Fraction and percentage of GC neurons excited or inhibited by the cross-modal cues.

observed in untrained animals (1.22 ± 0.06 and 1.44 ± 0.07 for untrained and trained respectively; t-test: $t_{(155)} = -2.02$; p<0.05) (**Figure 5C**).

In summary, learning to associate cross-modal cues with sucrose lead to an increase in inhibitory responses and to the development of responses characterized by a more dynamic firing modulations.

## Coding of cue-related information

To determine whether the distinct response dynamics observed in trained and untrained rats were associated with differences in coding of cue-related information we performed a decoding analysis (*Jones et al., 2007*; *Samuelsen et al., 2012*; *Jezzini et al., 2013*). This approach allowed us to investigate, in both untrained and trained rats, differences in how the GC ensemble activity encodes for the four cross-modal stimuli. The ensembles were composed of all the neurons recorded in a certain condition (untrained: n = 135 [**Figure 6A**]; trained: n = 118 [**Figure 6B**]). Briefly, each single trial of ensemble activity in response to a specific stimulus was classified by comparing it with the average ensemble responses for each of the four cross-modal stimuli. The time course of the classification performance was analyzed over 2.5 s following stimulus onset and was computed for each 200 ms bin.

In both conditions, the population of neurons recorded allowed for a significantly above chance classification of the sensory modality of cues. GC neurons were able to decode the different stimulus modalities both in untrained and trained rats (**Figure 6A–B**). The peak classification performance was observed 100 ms after stimulus onset and was 0.67 for neurons in untrained rats and 0.70 for neurons recorded in trained animals. The decoding performance decreased after the early peak, but continued to be significantly above chance for the duration of the cross-modal stimulation. In both conditions, classification remained significant in the 500 ms following the cue offset.

The results of this classification analysis were further analyzed to identify possible differences in decoding for trained and untrained subjects. For each cross-modal stimulus, the proportion of correct hits and errors was computed. Correct hits were trials in which a stimulus was correctly classified as belonging to a specific modality (for example, an air puff being classified as a somatosensory stimulus and not as an odor, tone or light; referred hereafter as correct hits). Errors were trials in

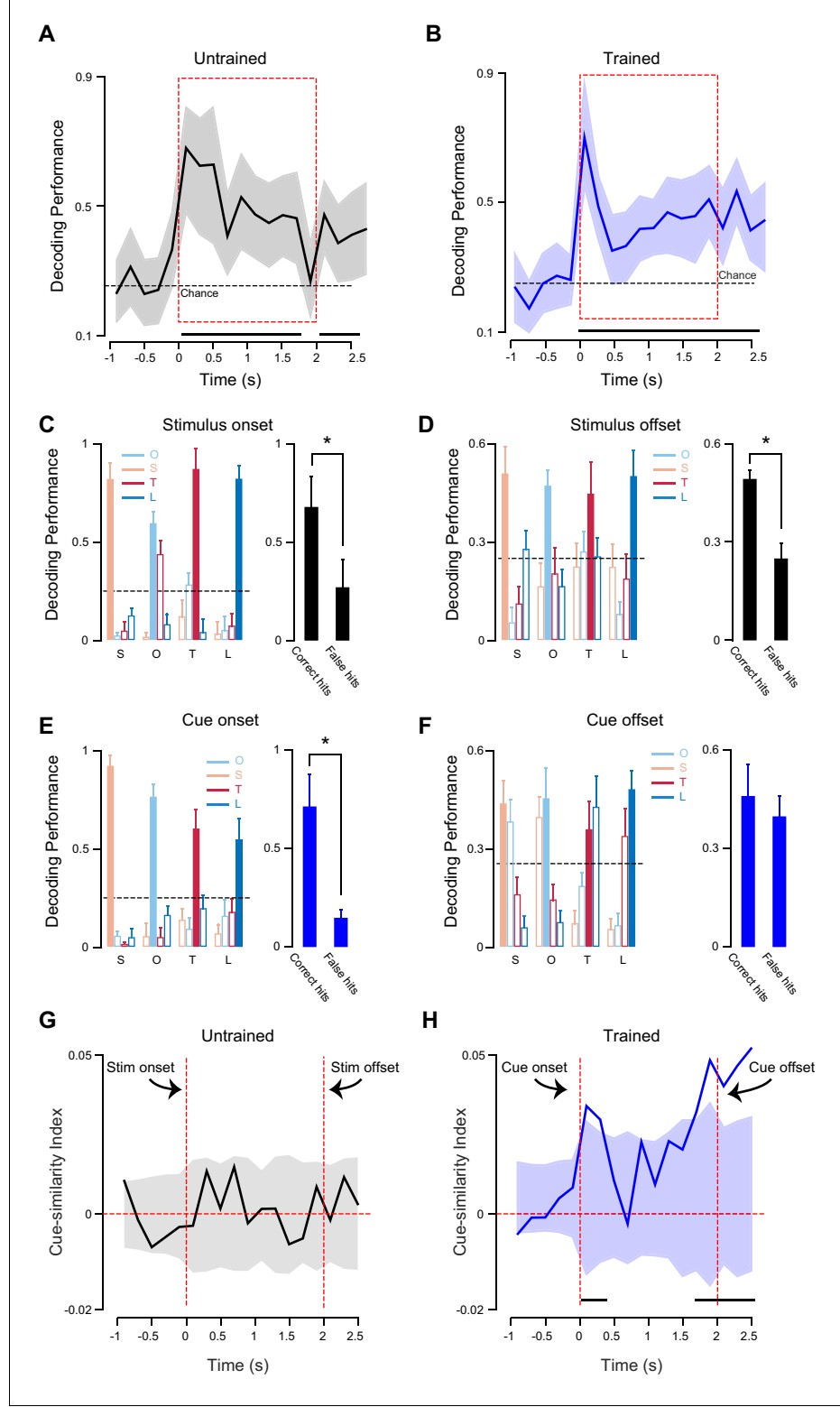

**Figure 6.** Coding of different sensory modalities for untrained and trained rats. (**A**) Time course of the classification performance for cross-modal stimuli in the group of untrained rats. (**B**) Time course of the classification performance for cross-modal anticipatory cues in the group of rats that underwent an associative learning. For A and B: solid line represents correctly classified trials (as% ) based on population activity; shading gray area around traces represents 95% bootstrapped CI. Thick black horizontal lines below traces indicate

*Figure 6 continued on next page*

*Figure 6 continued*

significance from the chance level (95% bootstrapped CI above chance). Dotted horizontal lines indicate chance performance. Red dotted boxes represent the period of cross-modal stimulus presentation. (**C**) Left most panel: histograms for average cross-modal stimuli classification performance for neurons recorded in untrained rats. Classification performance is evaluated for a 200 ms time bin after stimulus onset. Error bars indicate SD. Right most panel: average classification for correct hits and false hits (second highest classification values after correct hits). Error bars indicate SEM. (**D**) Left most panel: as in **C**, but for a 200 ms time bin after stimulus offset. (**E**) Left most panels: histograms showing cross-modal cues classification performance for neurons recorded in trained rats. Classification performance is evaluated for a 200 ms time bin after stimulus onset. Error bars indicate SD. Right most panel: average classification for correct hits and false hits (second highest classification values after correct ones). Error bars indicate SEM. (**F**) Left most panels: as in **E**, but 200 ms after stimulus offset. (**G**) The time course of the CI measured for 200 ms bins over 1 s before and 2.5 s after cue onset in untrained rats. (**H**) Time course of the CI measured for 200 ms bins over 1 s before and 2.5 s after cue onset in trained rats. For **C**, **D**, **E**, **F**: each group of bars shows the percentage of trials classified for each stimulus. Labels under each bar indicate the actual delivered stimulus (S for somatosensory, O for odor, T for tone and L for light). For **G** and **H**: solid lines represent correct CI (measured assuming somatosensory and odor cues being similar). Shading around traces represents 95% bootstrapped CI (where the similarity between cues is shuffled, for example tone-odor or light-somatosensory being similar). Black horizontal lines indicate bins in which the CI is significantly different from chance. Asterisk indicates in panels **C**, **D** and **E** indicate p<0.05 for t-test.

The following source data is available for figure 6:

**Source data 1.** Average and standard error of 'correct' and 'false' hits.

---

which a stimulus was misclassified (for example, an air puff being incorrectly classified as an odor, tone or light). In the case of errors, we focused on the most frequent misclassifications (for example, an air puff being more frequently misclassified as an odor compared to tone or light; referred hereafter as false hits). Percentages of correct and false hits over the total number of trials for each cross-modal stimulus were computed for two temporal windows: shortly after the onset of the cross-modal stimulus (from 0 to 200 ms) and shortly after stimulus offset (from 2.5 to 2.7 s). For untrained animals, neural activity led to qualitatively similar classification performance in both windows. After onset and after offset of cross-modal stimuli, the average of correct hits significantly outnumbered the average of false hits (*Figure 6C*; stimulus onset: $0.67 \pm 0.16$ and $0.26 \pm 0.16$ correct and false hits respectively; t-test: $t_{(6)} = 3.54$; p<0.05; *Figure 6D*; stimulus offset: $0.48 \pm 0.03$ and $0.25 \pm 0.04$ correct and false hits respectively; t-test: $t_{(6)} = 5.31$; p<0.05). For trained animals, performance in the two windows was qualitatively different. After cue onset, activity allowed for optimal coding of stimulus identity, with correct hits dramatically outnumbering false hits (*Figure 6E*; $0.70 \pm 0.17$ and $0.18 \pm 0.02$ correct and false hits respectively; t-test: $t_{(6)} = 5.89$; p<0.05). After the offset of the cue, i.e., in the delay interval (500 ms) preceding sucrose delivery, neural activity lead to an equally large number of correct and false hits (*Figure 6F*; $0.45 \pm 0.09$ and $0.38 \pm 0.04$ correct and false hits respectively; t-test: $t_{(6)} = 1.11$; p=0.30). It is important to notice that cross-modal stimuli were not randomly misclassified. Indeed the decoder was more likely to confound stimuli that shared similar associability. During the delay period, olfactory and somatosensory cues (the strongly associable cues, as determined by the faster learning rate in the behavioral experiment) could be easily confused with each other; similarly, visual and auditory cues (the two stimuli with weaker associability) could be misclassified. Misclassifications between cues with different associability were significantly less frequent (*Figure 6F*).

To further confirm this observation, a cue-similarity index (CI) was computed (*Figure 6G–H*). This index quantifies the similarity of firing rates evoked by stimuli that share associability (see Material and methods for details). Positive values of this index indicate that a neuron is modulated similarly by cues with similar taste-associability power (odor and somatosensory stimuli or tone and light), while negative values indicate the opposite (no similarity between responses to stimuli with similar associability). Comparison of the CIs computed for untrained and trained rats show differences in the time-course between the two conditions (*Figure 6G–H*). While before association GC neurons show no above chance peak in the CI throughout the entire time-course (*Figure 6G*), after learning GC neurons showed a significant peak in CI right after cue onset and a stronger and sustained peak

starting 1.7 s after cue onset and lasting for the duration of the delay (500 ms between cue offset and sucrose presentation) period (*Figure 6H*).

These results show that the sensory modality of non-gustatory stimuli is encoded throughout the stimulation period as well as following the offset of the stimulus in both untrained and trained rats. GC neurons in trained animals can also encode the associability of the cue during the delay period.

## Discussion

In this article, we present results unveiling how cross-modal stimuli are represented in a primary sensory cortex in the absence or in the presence of associative learning. Here we defined as 'cross-modal' neural responses evoked by sensory stimuli that do not belong to the modality typically represented in the primary sensory area of interest. In the case of GC, cross-modal responses are changes in firing rates evoked by auditory, visual, olfactory and somatosensory stimuli. While this definition does not distinguish between responses purely representing the sensory features (i.e., the modality) of the different stimuli and responses purely representing learned associations, our data provide evidences that both types of representations can exist in a primary sensory cortex.

We recorded single unit responses to olfactory, somatosensory, visual and auditory stimuli from GC of alert rats. In the first experiment, cross-modal stimuli were presented without establishing any association with tastants. We found that neurons in GC represent stimuli of all four modalities, but have a strong bias in favor of somatosensory and olfactory stimuli. Analysis of the convergence of different modalities onto the same neuron revealed that in untrained animals GC neurons are largely unimodal. This result implies that, in the absence of taste-association, representations of cross-modal stimuli in GC are non-overlapping. Cross-modal responses were mostly excitatory and tended to be tonic, lasting for the entire duration of the stimulation window and beyond. Decoding analysis revealed that this type of neural activity endowed GC with the ability to successfully recognize stimuli of different sensory modalities.

In the second experiment, we trained a new group of rats to associate cross-modal stimuli with the delivery of sucrose and compared their responses with those recorded in untrained animals. Training showed that different modalities lead to different learning rates, with olfactory and somatosensory stimuli (e.g., those more represented in GC neurons of untrained animals) becoming associated with sucrose more rapidly than auditory and visual stimuli. Analysis of neural responses after training revealed that associative learning was linked to a dramatic increase in the prevalence of neurons responding to non-gustatory stimuli compared to untrained animals. This increase led to an expansion of the contingent of neurons that responded to multiple stimuli. While in untrained animals only a few taste selective neurons responded to cross-modal stimuli, after associative learning the majority of taste selective neurons were also cue responsive. Similarly, as a result of the expansion of cross-modal representations, the number of neurons responding to more than one cue increased after associative learning. Analysis of taste responses in cue responsive neurons revealed no bias toward any specific tastant, demonstrating that cue-sucrose pairing did not result in a preferential representation of sucrose. Indeed, cue responsive neurons maintained the ability to represent multiple tastants. Quantification of the breadth of tuning revealed that learning reduced stimulus selectivity and increased the breadth of tuning in GC neurons. Learning also increased the number of neurons that were inhibited by cross-modal stimuli and changed the dynamics of the responses. Instead of being tonic and monophasic as in untrained animals, cross-modal responses in trained rats showed multiple modulations lasting for the entire duration of the cue and beyond. In addition, decoding analysis revealed that cue responses after learning retain information about the sensory modality of the cue; in addition, they acquire information regarding the associability of different modalities.

Altogether, these results provide evidence suggesting that associative learning can modify the representation of cross-modal stimuli in primary sensory cortices.

### Cross-modal integration in the gustatory cortex

Cross-modality has been demonstrated in multiple sensory areas (*Katz et al., 2001*; *Gottfried et al., 2002*; *Kayser and Logothetis, 2007*; *Iurilli et al., 2012*; *Olcese et al., 2013*; *Liu et al., 2015*; *Ibrahim et al., 2016*). In this study, we investigate cross-modality in the gustatory cortex. Taste is a good system to investigate how primary sensory cortices integrate cross-modal

information (*Dalton et al., 2000*; *Katz et al., 2001*; *Simon et al., 2006*; *de Araujo and Simon, 2009*; *Escanilla et al., 2015*). The experience of eating is inherently multimodal, involving the integration of gustatory, somatosensory and olfactory signals in a single percept called 'flavor' (*Katz et al., 2001*; *De Araujo and Rolls, 2004*; *Spence, 2015*). In addition, the approach to food is guided by multisensory cues that predict its availability (*Gardner and Fontanini, 2014*; *Kusumoto-Yoshida et al., 2015*).

Here we chose to focus on cross-modality related to anticipation because this is a phenomenon that generalizes to other sensory areas (*Freeman, 1983*; *Schoenbaum and Eichenbaum, 1995*; *Meftah el et al., 2002* ; *Shuler and Bear, 2006*) As a result, we deliberately avoided stimuli typically linked to flavor, such as intraoral tactile stimulation and retro-nasal olfactory stimulation (*Yamamoto et al., 1988*; *Simon et al., 2006*; *Gautam and Verhagen, 2012*). Instead, we selected extra-oral, cross-modal stimuli similar to those that are typically used in behavioral and sensory experiments (*Brecht and Sakmann, 2002*; *Derdikman et al., 2003*; *Barnes et al., 2008*; *Wesson et al., 2009*; *Nonkes et al., 2012*; *Ollerenshaw et al., 2012b*; *Samuelsen et al., 2012*; *Centanni et al., 2013*; *Devore et al., 2013*; *Headley and Weinberger, 2013*; *Gardner and Fontanini, 2014*).

Several studies have explored responses to cross-modal stimuli in GC of anesthetized (*Yamamoto et al., 1981*; *Rodgers et al., 2008*) or alert, untrained animals (*Yamamoto et al., 1989*; *Katz et al., 2001*). Recently it was shown that learning the taste-predictive value of an auditory cue increases the prevalence of auditory responses in GC (*Samuelsen et al., 2012*; *Gardner and Fontanini, 2014*). So far, however, no study has assessed GC responses to cross-modal stimuli in both untrained and trained animals. Our experiments show that GC can encode stimuli of all sensory modalities, even stimuli that have relatively little ethological relationship to taste, like sounds or flashes of light. In untrained animals, each sensory modality engages a selective group of unimodal neurons, endowing GC with the ability to discriminate different stimuli. This result indicates that cross-modal responses in GC are not just reflecting the general processing of non-gustatory information. If cross-modal responses simply encoded the presence of generally arousing (*Talsma et al., 2010*), rewarding (*Shuler and Bear, 2006*; *Pooresmaeili et al., 2014*) or distracting (*Iurilli et al., 2012*) stimuli, they would involve overlapping sets of neurons for each stimulus. The presence of cross-modal responses in untrained animals may reflect GC's intrinsic tuning toward multisensoriality and its ability to encode different dimensions linked to food expectation. These responses can also represent a substrate upon which learning can operate (*Vogels et al., 2011*). Indeed, pairing stimuli with sucrose leads to an expansion of cross-modal representations and the development of cue responses. Responses become temporally more complex relative to those observed in untrained animals, probably due to GC inputs from multiple regions (*Samuelsen et al., 2012*, *2013*; *Liu and Fontanini, 2015*). After learning, fewer neurons are unimodal and more become multimodal, suggesting that GC is now also encoding for a common property of the cross-modal stimuli, e.g., their taste-anticipatory value. In addition to encoding expectation, decoding analysis indicates that, after associative learning, GC can encode both the identity of different cross-modal stimuli, and their associability. These results confirm and significantly expand data from the literature on the functional role of GC (*Samuelsen et al., 2012*; *Gardner and Fontanini, 2014*).

Analysis of the distribution of responses to each stimulus reveals a key feature of cross-modal responses in GC: their bias toward stimuli that are more promptly associated with gustatory rewards (e.g., odors and whisker stimulation). We cannot exclude that both neural and behavioral biases might reflect differences in detectability or saliency across stimuli sensory modalities before training. Indeed, it is possible that, before learning, the air puff and the odor may be more intense than light and tone. In this regard, it would be interesting for future studies to explore various stimulus intensities and establish the intensity-response relationship of cross-modal responses. We would not be surprised if varying the intensity of stimuli affected the number of neurons recruited, however, we believe that the bias for odors and somatosensory stimuli would still persist. This belief is supported by the strong interconnectivity between GC with olfactory (*Shipley and Geinisman, 1984*) and somatosensory areas (*Guldin and Markowitsch, 1983*).

While differences in cross-modal stimulus intensity and detectability could partially explain the observed bias before learning, such an interpretation does not apply to the bias observed after the cue-sucrose association. After 14 days of training, behavioral responses for all the cues become similar, indicating that the four cues acquire similar saliency. Despite the absence of a behavioral bias,

the neural bias was maintained, suggesting that it cannot be fully accounted by differences in behavioral saliency among stimuli. One possible explanation for the persistence of the bias after learning may involve the non-uniform number of sessions at final performance level for the different cues. Indeed, rapid learning for somatosensory and olfactory cues resulted in a larger number of above-criterion sessions compared to visual and auditory cues. To investigate this possibility, we matched the number of sessions at final performance level across cues and quantified the fraction of neurons responsive to the four cross-modal stimuli. The bias still persisted after matching the number of sessions, demonstrating that it is inherent to GC and does not depend on the amount of cue-sucrose exposure.

In summary, these results suggest that the imbalance in responsiveness to the different cues might reflect, at least in part, the strong ecological relationship between odors, somatosensory stimuli and taste in rodents.

## Cross-modal representations and associative learning in cortical areas

Responses to cross-modal stimuli have been extensively studied in the superior colliculus and associative cortices (*Stein and Arigbede, 1972*; *Meredith and Stein, 1986*; *Lipton et al., 1999*; *Fuster et al., 2000*). This body of literature has demonstrated fundamental principles of how cross-modality is represented and is shaped by experience and learning (*Watanabe, 1992*; *Lipton et al., 1999*; *Fuster et al., 2000*; *Yu et al., 2010*; *Xu et al., 2015*). Alas, our knowledge of cross-modal processing in primary sensory cortices is nowhere near that level of sophistication.

Classic results from extracellular recordings in somatosensory, visual and auditory cortices of anesthetized rats demonstrated that cross-modal responses exist mostly in transitional zones between sensory areas (*Wallace et al., 2004*; *Olcese et al., 2013*). Recent work from alert animals, however, has shown that cross-modal responses are not limited to border zones (*Lakatos et al., 2007*; *Iurilli et al., 2012*; *Maier et al., 2012*). Multimodality in primary sensory cortical areas has been mostly studied with a focus on bimodality (*Driver and Noesselt, 2008*); to our knowledge only a handful of studies have investigated how single neurons in a primary sensory cortex respond to stimuli of more than three modalities (*Yamamoto et al., 1988*, *1989*). As a result, our understanding of the convergence of information from multiple modalities on individual sensory cortical neurons is limited. Our data address this important issue directly and show that in alert, untrained rats, GC can respond to stimuli of all five sensory modalities, and that the majority of neurons respond to a single modality. The degree of convergence of cross-modal information on single neurons, however, is not fixed; indeed our data demonstrate that it can be modified by associative learning.

Associative learning is fundamental for shaping cross-modal responses in frontal cortices (*Watanabe, 1992*; *Lipton et al., 1999*; *Fuster et al., 2000*), and in subcortical areas (*Yu et al., 2010*; *Xu et al., 2015*). Associative learning is also known to modify sensory representations in primary sensory cortices (*Zhou and Fuster, 2000*; *Brosch et al., 2005*; *Shuler and Bear, 2006*; *Weinberger, 2007*; *Grossman et al., 2008*; *Chen et al., 2011*), albeit this body of literature focused exclusively on few single modalities. Indeed, little is known about how learning affects cross-modal representations in primary sensory cortices. Our data demonstrate that associative learning can expand the pool of neurons responding to cross-modal stimuli and enhance the ability of neurons to represent multiple modalities. It is worth mentioning that, the set of cross-modal stimuli used in this study is not homogeneous. Some stimuli, those with stronger associability, recruit many more GC neurons than others. Nevertheless, learning is capable of expanding the representations for all the stimuli, regardless of how much they activate GC at baseline. In addition to expanding cross-modal representations, associative learning enhances the similarity of responses to stimuli with similar associability.

The results from trained rats strongly indicate that GC's cross-modal representations are plastic and, hence, likely to be affected also by conditioning with aversive gustatory outcomes (quinine) or by extinction. Previous work demonstrated that using two gustatory outcomes that differ in palatability (sucrose, palatable, and quinine, aversive) could lead to the genesis of cue responses representing specific expectations (*Gardner and Fontanini, 2014*). On the basis of that result, we can speculate that the presence of aversive outcomes might drive the expansion of cross-modal representations as much as rewards. In addition, the presence of different outcomes might be important in inducing the development of cue responses showing a bias for specific tastants. As for extinction, findings from the same study (*Gardner and Fontanini, 2014*) suggest that it might shrink cross-

modal representations. It is worth mentioning that the results in (*Gardner and Fontanini, 2014*) also showed that neurons in GC could change their firing rates in trials in which sucrose was predicted by a cross-modal cue but omitted. This phenomenon, which was not further investigated in the present paper, demonstrates that cross-modal activity in GC may also reflect learned expectations of reward timing – a phenomenon well studied in the primary visual cortex (*Shuler and Bear, 2006*; *Liu et al., 2015*). Future studies will rely on omission trials as well as sham trials (i.e., trials in which neither a cue nor sucrose is presented, but that have the same ITI as any other trial) to investigate the effects of expectations of stimulus and reward timing on neural activity and mouth movements.

In conclusion, our experiments demonstrate the importance of learning in shaping the representation of cross-modal information in primary sensory areas. These results further emphasize the importance of considering experience, learned associations and expectations when interpreting cross-modal responses in sensory cortices.

## Materials and methods

### Experimental subjects

The experiments in this study were performed on ten female Long-Evans rats (250–350 g; Charles River). We chose female rats for their amenability to experiments in restraint and for consistency with previously published work (*Samuelsen et al., 2012*; *Jezzini et al., 2013*; *Samuelsen et al., 2013*; *Gardner and Fontanini, 2014*; *Liu and Fontanini, 2015*). The stages of the estrous were not monitored. However, since the neural activity was recorded for 14 consecutive days (in both groups of trained and untrained rats), and since the estrous cycle lasts 4–5 days, data were likely collected across all rats' estrous stages. Rats were individually housed and maintained on a 12 hr light/dark cycle with *ad libitum* access to food and water unless otherwise specified. All experimental procedures were approved by the Institutional Animal Care and Use Committee at Stony Brook University and complied with university, state, and federal regulation on the care and use of laboratory animals.

### Surgery

Rats were anesthetized with an intraperitoneal injection of a mixture of ketamine/xylazine/acepromazine (100, 5.2 and 1 mg/kg, respectively). Supplemental doses of anesthetic (30% of initial dose) were administered when needed. Rats' body temperature was maintained at 37°C throughout the surgery. Once fully anesthetized, rats were placed on a stereotaxic device, their scalp was scrubbed with iodine and excised to expose the skull. Holes were drilled for electrodes above GC (antero-posterior: +1.4 mm, medio-lateral: ± 5 mm from bregma) and for anchoring screws in seven other positions (*Jezzini et al., 2013*; *Gardner and Fontanini, 2014*). Electrodes consisted of drivable bundles of 16 individual formvar-coated nichrome micro-wires (*Samuelsen et al., 2012*, *2013*), which were bilaterally inserted above GC (dorso-ventral: −4 mm from dura). Movable bundles were lowered ~200 µm after each experimental session allowing us to record multiple ensembles in the same animal. After insertion of the electrodes bundles, intra-oral cannulae (IOCs) were bilaterally inserted to allow for the delivery of gustatory stimuli directly into the oral cavity. Electrode bundles, IOCs and a head bolt (for the purpose of head restraint) were cemented to the skull with dental acrylic. Animals were allowed to recover for a minimum of 7 days before training began. The proper placement of electrodes was verified, at the end of each experiment by means of standard histological procedures (*Samuelsen et al., 2012*; *Jezzini et al., 2013*; *Gardner and Fontanini, 2014*; *Liu and Fontanini, 2015*) (see below).

### Restraint training and behavioral paradigm

Following recovery from the surgery, rats from untrained and trained groups were placed on a water restriction regime (30–45 min of daily access to water). After 2–3 days, rats from both groups began training to sit calmly in restraint while receiving deliveries of water (~50 µl of water with a variable inter-trial interval [ITI]) via IOCs. Session duration was progressively increased to 1 hr and 10 min, over 2 to 3 weeks. Animal selection (weight, age and gender), housing, surgical procedures, post-surgical recovery, water restriction, restraint training and IOC training were the same ('common' phase) for both groups of animals used in this study. Procedures differed only following the common

phase; the group of trained rats (n = 5; referred as *Trained* in the main text and figures) underwent associative leaning, while the group of untrained rats (n = 5; referred as *Untrained* in main text and figures) entered the experimental phase.

## Untrained rats

In the experimental phase for this group of animals, we recorded spiking activity and orofacial movements evoked by random delivery of the following cross-modal and gustatory stimuli: an odor, a puff of air on the distal portion of the rats' whiskers (*air puff*), a light, a tone, unexpected sucrose, unexpected NaCl, unexpected citric acid and unexpected quinine (see Stimuli delivery section below for further information). After each stimulus delivery, with a variable interval (25 ± 5 s), ~50 µl of water (*Rinse*) was delivered. 25 ± 5 s after a rinse a new trial began (see *Figure 2—figure supplement 1*). The ITI was 50 ± 10 s in order to avoid associations between cross-modal, 'non-gustatory' stimuli and tastants. Each stimulus was presented from 8 to 14 trials. Recordings were performed for 14 days (one session per day); at the end of each daily session bundles of electrodes were lowered ~150 µm.

## Trained rats

This group of animals was trained to associate four cross-modal, 'non-gustatory' stimuli (the conditioned stimulus; an air puff on the distal whiskers, an odor, a tone or a light stimulus; see Stimuli delivery section below for further information) with a gustatory stimulus (the unconditioned stimulus; sucrose) in a Pavlovian task. Cross-modal stimuli were presented in a pseudorandom fashion; they always lasted 2 s and their onset preceded the delivery of sucrose by 2.5 s. Trials occurred at a 50 ± 10 s interval (see *Figure 3—figure supplement 1*). The entire conditioning period lasted 14 days (see *Figure 2*; one session per day). In each daily session, rats received 20 trials of each cue-taste pair. The time-course of classical conditioning was monitored every day by quantifying conditioned mouth movements (see Mouth movement analysissection). After 14 days of training, rats entered the recording phase. In this phase, we recorded spiking activity and orofacial movements evoked by random delivery of the following cross-modal and gustatory stimuli: an odor, a puff of air on the distal portion of the rats' whiskers (*air puff*), a light, a tone, expected and unexpected sucrose, unexpected NaCl, unexpected citric acid and unexpected quinine (see Stimuli delivery section below for further information). In this experiment, as during the conditioning phase (see above), each cross-modal cue anticipates by 2.5 s the delivery of sucrose (expected sucrose), whereas unexpected sucrose and the other three tastants are delivered without cross-modal stimuli preceding them (see *Figure 3—figure supplement 1*). 25 ± 5 s after each stimulus delivery ~50 µl of water (*Rinse*) was delivered. 25 ± 5 s after a rinse a new trial began (see *Figure 3—figure supplement 1*). The ITI was 50 ± 10 s. Between 8 and 14 trials for each stimulus were delivered. Recordings were performed for 14 days (one session per day); at the end of each daily session bundles of electrodes were lowered 150 µm.

## Stimuli identity and delivery

Gustatory stimuli were delivered through a manifold made of multiple, fine polyimide tubes slid into the IOC (*Fontanini et al., 2009*; *Samuelsen et al., 2012*). A pressurized taste delivery system delivered tastants through the IOCs in aliquots of 40 µl with a 25–30 ms pulse (*Gardner and Fontanini, 2014*). A rinse of water (50 µl) followed the taste after 25 ± 5 s in order to wash the oral cavity. The concentrations of the four basic tastants used were 0.1 M for sucrose, citric acid and NaCl, while a concentration of 0.001 M was used for quinine-HCl. All the tastants were from Sigma-Aldrich. The auditory cue consisted of a 75 dB, 2 kHz single tone. Two green LEDs with a peak emission wavelength of 575 nm delivered the visual cue. The intensity of the light was 0.147 lm. The LEDs were placed bilaterally at a distance of ~5 cm from each eye. The somatosensory cue consisted of a bilateral air puff generated by a pico-spritzer (~5 psi) delivered from two PFA tubes (5 mm diameter). The outlet of the tube was placed ~5 cm from the rats' whiskers. The air puff was directed to the distal portion of the whiskers. The olfactory cue (0.1% Isoamyl acetate) was delivered via a custom-built, computer-controlled olfactometer (*Verhagen et al., 2007*), and delivered at a ~2 cm distance from the rats' nose. The duration of all cross-modal sensory stimuli was 2 s. The choice of stimulus identity and intensity was consistent with existing studies showing that rats can easily detect the stimuli we used. Puffs of air on a single or on multiple whiskers have been used as either cues or sensory stimuli

in multiple studies (*Brecht and Sakmann, 2002*; *Derdikman et al., 2003*; *Civillico and Contreras, 2006*; *Ollerenshaw et al., 2012a*). Green light cues are typically used in behavioral studies as conditioned stimuli (*Jacobs, 2009*; *Nonkes et al., 2012*; *Headley and Weinberger, 2013*). Auditory stimuli with a frequency and intensity similar to the ones used here have been used to condition rats in prior studies from our laboratory (*Samuelsen et al., 2012*; *Gardner and Fontanini, 2014*) and as stimuli to be discriminated (*Centanni et al., 2013*; *Xiong et al., 2015*). Finally, several studies demonstrate that rats can easily detect and discriminate the odor Isoamyl acetate when presented at the concentration equal or lower to the one (0.1%) used in this study (*Barnes et al., 2008*; *Wesson et al., 2009*; *Devore et al., 2013*). While all the stimuli that we relied on can be easily detected, no information is available on their relative saliency in naïve animals.

## Electrophysiological recording and analyses

Signals were amplified, band pass filtered (300–8000 Hz), digitized and recorded (Plexon [Dallas]; sampling rate: 40 KHz). Single units were off-line sorted using a template algorithm, cluster-cutting techniques and examination of inter-spike interval plots (Offline Sorter, Plexon). The average number of units recorded per rat was 27 ± 5.2 and 23.6 ± 2.5 for the *Untrained* and *Trained* groups, respectively. Data were analyzed with custom-written scripts in Matlab (MathWorks, Inc., Natick, Ma). Single-neuron and population peristimulus time histograms (PSTHs) were extracted for cross-modal stimuli and taste deliveries. A bin size of 200 ms was used unless otherwise specified. Single units that displayed a large power-spectrum peak around 6–10 Hz were considered 'somatosensory' (*Katz et al., 2001*; *Jezzini et al., 2013*) and excluded from additional analysis.

### Stimulus responsiveness

Responsiveness to stimuli was assessed using a 'change point' (CP) analysis as in Jezzini et al., (*Jezzini et al., 2013*) and Liu and Fontanini (*Liu and Fontanini, 2015*). We first computed the cumulative distribution function (CDF) of spike occurrences across all trials for a given stimulus (*Jezzini et al., 2013*). For each neuron and for each stimulus we analyzed the CDF in the time interval starting 1 s before and ending 2 s (cross-modal stimuli) or 2.5 s (taste stimuli) after stimulus delivery. Sudden changes in firing rates resulted in a piecewise change in the CDF slope and in the identification of a CP. If no CP was detected for any of the stimuli, the neuron was deemed not responsive. By definition, a neuron was stimulus responsive if at least one significant CP was found after stimulus presentation. The CP analysis was also used to establish the sign of response, i.e., excitation or inhibition, and the average number of modulations (data shown in *Figure 5B* and *Figure 5C*, respectively). For the sign of the response only the first change (first CP) was considered. For the number of modulations all CPs detected in 2.5 s after stimulus were quantified.

### Taste selectivity

A neuron was considered taste selective if it was responsive to at least one of the four tastes and its response varied significantly across different tastants in either magnitude or time course. The latter was defined with two-way ANOVA [taste identity, time course], using 200 ms bins in the 0 to 2.5 s post-delivery intervals. A neuron was defined taste selective if either the taste main effect or interaction of the two effects (taste identity X time course) was found significant.

### Significance of convergence between responses

A bootstrap analysis was used to determine if the fraction of neurons responding to more than one cross-modal stimulus (or being both taste selective and cross-modal responsive) was significantly different from what expected from the joint probabilities of responses to individual stimuli. The null hypothesis is that responses to different stimuli are independent from each other. Briefly, for the case of cross-modal responses we simulated 10,000 combinations of neurons each with the experimentally observed probability of responding to air puff (p(air puff)), odor (p(odor)), tone (p(tone)), and light (p(light)). Before learning, the experimentally observed parameters were as follows: number of neurons = 135; p(air puff) = 0.15; p(odor) = 0.16; p(tone) = 0.04; p(light) = 0.03. After learning: number of neurons = 118; p(air puff) = 0.34; p(odor) = 0.33; p(tone) = 0.16; p(light) = 0.11. For the case of the convergence between taste selectivity and cross-modal responsiveness we simulated 10,000 combinations of neurons with the following experimentally observed probabilities. Before

learning: n = 135; the probability of being taste selective = 0.64; the probability of being cross-modal responsive = 0.33. After learning: n = 118; the probability of being taste selective = 0.57; the probability of being cross-modal responsive = 0.58. The simulations were used to build a distribution of expected probabilities for combined responses, which allowed for the determination of a confidence interval and a significance threshold (p<0.05) for comparison with the experimentally observed data.

## Decoding analysis

To investigate the time course of cross-modal stimuli coding in GC before and after associative learning, we relied on a stimulus classification procedure applied at each 200 ms time bin. Details of this procedure can be found in *Jezzini et al. (2013)* and *Liu and Fontanini (2015)*. This method determines how well a given cross-modal stimulus is encoded by a population of neurons in a given temporal bin and is grounded on both a cross-validation procedure as well as a Euclidean distance-based classifier. To compute a population decoding analysis of cross-modal stimuli on both untrained and trained rats, we constructed a 'pseudo-population' of neurons, collecting units from different sessions and rats under the same stimulus condition. Each cross-modal stimulus was denoted as a point in a 2 dimensional vector space (R x C) of firing rates, where R is the number of units included in the pseudo-population (only units from sessions with at least 8 trials for each cross-modal stimulus were considered), and C is the number of temporal bins. To assess the statistical significance of the decoding analysis (confidence interval, CI), we used a bootstrap procedure in which we sampled with replacement a subset of 80% of the neurons from the whole population for each bootstrap run (100 bootstrap runs were used). Significant decoding was defined by the lower bound of 95% bootstrapped confidence interval (CI) larger than chance level (0.25).

## Area under the receiver operating characteristic curve (auROC) method

To compare GC neurons response profile to cross-modal stimuli before and after associative learning (data shown in *Figure 5*) and to compute the cue-similarity index (see below) we used the auROC method for normalizing PSTHs (200 ms bin). The detailed description of this method can be found here (*Cohen et al., 2012*; *Gardner and Fontanini, 2014*). Briefly, the auROC method compares two distributions of firing rates before and after stimulus presentation, i.e., baseline vs. evoked. Values larger than 0.5 express the likelihood that the firing rate in the bin is above baseline, whereas values below 0.5 express the likelihood that the firing rate in the bin is below baseline.

## Selectivity index

To evaluate the stimulus selectivity (four tastants and four cross-modal stimuli), a sharpness index was computed, based on mean firing rates from the post-stimulus epoch (2.5 s) (*Rainer et al., 1998*; *Yoshida and Katz, 2011*). Sharpness was defined as $(n - \Sigma \, FRi/FRbest)/(n - 1)$, where FRi is the mean firing rate for each stimulus (i = 1–8), FRbest is the maximum firing rate among stimuli, and n is the total number of stimuli (n = 8). A sharpness index of 1 indicates that a neuron responded to one stimulus, and the value 0 indicates equal responses across stimuli.

## Cue-similarity index

The cue similarity index (CI) was computed as follows. As a first step, for each neuron we averaged the absolute value of the log-likelihood ratio of responses to cues with similar associability (e.g., odor, O, and whisker stimulation, S or Tone, T, and Light, L; $|R|_{(Similar)} = 0.5*(|\ln O/S| + |\ln T/L|)$ and to cues with different associability ($|R|_{(Dissimilar)} = 0.25*(|\ln O/T| + |\ln O/L| + |\ln S/T| + |\ln S/L|)$). O,S,T and L denote the value of the auROC-normalized response for odor, air puff, tone and light trials respectively. As a second step, we defined the CI as the differences between the two averages: CI = $|R|_{(Dissimilar)} - |R|_{(Similar)}$. Positive CI values indicate that a neuron responds similarly to cues with similar associability, while negative values indicate a higher similarity for responses to stimuli with different associability. In order to evaluate statistical significance, we shuffled the pairs (e.g., using O and T or L and S as pairs having similar associability) and extract the 95% bootstrapped CI (represented as shading areas in *Figure 6 G–H*).

## Mouth movement analysis

Orofacial movements were recorded at a rate of 30 frames per second with a camera placed underneath the rat's mouth. Images were acquired and synchronized with electrophysiological signals (Cineplex; Plexon) and off-line imported in Matlab (MathWorks, Inc., Natick, Ma) for analysis. Mouth movements were assessed by automated frame-by-frame video analysis (*Gardner and Fontanini, 2014*). A region of interest (ROI) for the rat's orofacial region was selected. We then computed the absolute difference of the average pixel intensity of the ROI across consecutive frames (referred as Δ pixel intensity in *Figures 1–3*). These values were normalized to background pixel intensity (as obtained from a second region of interest selected away from the orofacial region) to correct for changes due to fluctuations in background light intensity. This procedure allowed us to have a continuous (bin = 33 ms) record of mouth movements (see *Figure 2—figure supplement 1*). To monitor conditioning, mouth movements were averaged across trial types and analyzed in three temporal epochs: before stimulus onset (1 s before; referred in the text and *Figure 2—figure supplement 1* as Spontaneous epoch), during stimulus (from 0.5 to 1.5 s following stimulus onset; referred in the text and *Figure 2—figure supplement 1* as Cue epoch) and after sucrose presentation (from 2.7 to 3.7 s after cue onset; referred in the text and *Figure 2—figure supplement 1* as Taste epoch). For data shown in *Figure 2*, to avoid differences in the magnitude of orofacial movement recorded from different rats, single mouth movement trials were also normalized to the maximum value recorded in the 'taste' epoch interval analyzed. Trials in which the rat's mouth was already in motion or in which view of the orofacial region was obstructed were not included in the analysis.

## Histological procedures

At the end of the experiment, rats were terminally anesthetized using an intraperitoneal injection of a mixture of ketamine/xylazine/acepromazine and electrolytic lesions were made to several electrode wires to mark recording sites (7 µA cathodal current for 7 s). Subjects were then perfused through the left cardiac ventricle with saline followed by 10% formalin. Brains were sectioned into 80 µm coronal slices and standard histological procedures (cresyl violet or Nissl staining) were performed to track electrode locations.

## Data and statistical analysis

No explicit power analysis was used to compute the appropriate sample size. However, number of animals, number of neurons and number of trials are consistent with the majority of sensory electrophysiology studies. All analyses of electrophysiological signals and orofaciall movements were performed using custom Matlab (MathWorks, Inc., Natick, Ma) scripts and *R* software (*R Core Team, 2015*).

## Acknowledgements

The authors would like to acknowledge Dr. Arianna Maffei, Dr. Guillermo Esber, Dr. Giancarlo La Camera, Dr. Luca Mazzucato, Dr. Roberta Tatti, Martha Stone and Maffei's laboratory for their feedback and insightful comments. The authors would also like to acknowledge Dr. Matthew Gardner for insightful discussion during the initial phase of the project and Amy Cheung for histology. This work has been supported by Swiss National Science Foundation Fellowships P2GEP3_151816 and P300PA_161021 to RV and by National Institute on Deafness and Other Communication Disorders Grant R01-DC010389 to AF.

## Additional information

### Funding

| Funder | Grant reference number | Author |
| --- | --- | --- |
| Schweizerischer Nationalfonds zur Förderung der Wissenschaftlichen Forschung | P2GEP3_151816 | Roberto Vincis |

| | | |
|---|---|---|
| Schweizerischer Nationalfonds zur Förderung der Wissenschaftlichen Forschung | P300PA_161021 | Roberto Vincis |
| National Institute on Deafness and Other Communication Disorders | R01-DC010389 | Alfredo Fontanini |

The funders had no role in study design, data collection and interpretation, or the decision to submit the work for publication.

## Author contributions

RV, Conception and design, Acquisition of data, Analysis and interpretation of data, Drafting or revising the article; AF, Conception and design, Drafting or revising the article, Contributed unpublished essential data or reagents

## Author ORCIDs

Roberto Vincis, http://orcid.org/0000-0002-5812-7624
Alfredo Fontanini, http://orcid.org/0000-0003-4561-9563

## Ethics

Animal experimentation: All experimental procedures were performed according to approved Institutional Animal Care and Use Committee protocols (#244930-1) at Stony Brook University, and complied with university, state, and federal regulation on the care and use of laboratory animals.

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
