## [Decision Letter]

Thank you for submitting your article "Associative learning changes cross-modal representations in the gustatory cortex" for consideration by *eLife*. Your article has been reviewed by two peer reviewers, Geoffrey Schoenbaum (Reviewer #2) and Marshall G Hussain Shuler (Reviewer #3), and the evaluation has been overseen by a Reviewing Editor and Sabine Kastner as the Senior Editor.

The reviewers have discussed the reviews with one another and the Reviewing Editor has drafted this decision to help you prepare a revised submission.

Summary:

This is an exciting paper that describes "cross-modal" responses (i.e. neural responses to stimuli that are not directly related to a tastant), and their emergence with associative learning, in gustatory cortex. The authors described responses to stimuli (specifically, an air puff, an odor, a light stimulus, and a tone) from varying modalities (somatosensory, olfaction, vision, and audition) when these stimuli were either paired or unpaired with a subsequent delivery of sucrose. As the animals learned this association, (as measured by anticipatory licking), neurons in gustatory cortex (GC) begin to respond to these non-taste stimuli in greater numbers. The authors' claims are supported by the data and the analyses performed. One strength of this work is that the authors presented trained rats with a battery of tastes to show that an increase in cross-modal responses is likely not due to a general increase in responsiveness. Further, they go to efforts to demonstrate that learned orofacial behaviors in anticipation of sucrose are not distinguishable following stimuli in the learned state, whereas neural response fractions and form differ. And while cross-modal stimuli are quite distinguishable early in training at the beginning and end of the stimulus delivery, in the learned state, responses across the ensemble evolve in such a way that stimuli with similar associability become self-similar.

Thus, in summary, after conditioning, there was more activity associated with cross modal cues, with many GC neurons responding to two or more cues, thus increasing the overlap of neural representations with similar associability. The study is an elegant demonstration of the effects of learning on the activity in a primary sensory area.

Essential revisions:

One very minor interpretation issue regards the meaning of fractions of cells exhibiting responses to cross-model stimuli: as the saliency of the stimuli may not be equal across the set, differences in the speed of acquisition may be accounted for on that basis (a possibility identified by the authors). It may be of some interest to match the number of sessions at the final performance level to determine whether the fractions of cells responsive to the cross-modal stimuli are then equal (i.e., is the fraction a property of the number of sessions spent at final performance level?).

The authors could do a better job in the Results section of the text of explaining their criteria for a cue evoked response and for saying that a neuron did or did not fire to particular cues, as well as how associability was determined. For example, the examples in Figure 1 are very clear, but the criteria and rationale should be described clearly in the main text. Please spell out, as is done for some of the later analyses.

The authors note in the Discussion (subsection “Cross-modal integration in the gustatory cortex”) that the GC may be an ideal system to look for cross-modal coding. Leaving aside my question above about this, does this suggest that the findings here may not generalize to other primary association regions? Visual cortex? What about the olfactory system? I believe there are studies showing a variety of responses after training in olfactory bulb and piriform cortex to non-olfactory cues, as well as the visual cortex. This should be expanded on in the Discussion.

---

## [Author Response]

*Essential revisions:*

*One very minor interpretation issue regards the meaning of fractions of cells exhibiting responses to cross-model stimuli: as the saliency of the stimuli may not be equal across the set, differences in the speed of acquisition may be accounted for on that basis (a possibility identified by the authors). It may be of some interest to match the number of sessions at the final performance level to determine whether the fractions of cells responsive to the cross-modal stimuli are then equal (i.e., is the fraction a property of the number of sessions spent at final performance level?).*

Following the reviewer’s suggestion, we performed two additional analyses aimed at investigating if the fractions of cross-modal neurons varied depending on the number of sessions at final performance level. These new analyses are shown in a new figure (Figure 3—figure supplement 2) and in additional text (see “Representation of cues after associative learning” section of the Results).

The first analysis compares the fractions of cells responsive to cross-modal stimuli for sessions recorded on the same number of days after reaching final performance level. As described in the main text and the Methods section of the manuscript, our data set originates from recordings performed from day 15 (included) to 21 (included) after the beginning of cue-sucrose conditioning (referred as day 1). As shown in Figure 2, cue-sucrose associations occurred at day 3 for olfactory and somatosensory cues, and day 7 for auditory and visual cues. In order to evaluate the fraction of cue-responding GC neurons after matching the number of recording sessions at final behavioral performance, we analyzed sessions on days 15, 16, 17 for olfactory and somatosensory stimuli and sessions on days 19, 20, 21 for auditory and visual stimuli. Analysis on this smaller set of data confirmed the results presented for the entire dataset. We observed similar fractions of cue responsive neurons, with more GC neurons responding to odor and somatosensory compared to sound and light stimuli.

To further investigate the point raised by the reviewer, we also computed the distribution of the number of cross-modal responsive neuron across recording days (from day 15 to day 21). If the fraction of GC neurons responsive to cross-modal stimuli depends on the familiarity with learned stimuli, we should observe a gradual increase across days. However, the number of cross-modal neurons responsive to each cue remained constant, with a slight general decrease over time as highlighted by the linear fit (Figure 3—figure supplement 2).

Taken together, these new analyses confirm that the bias observed in the number of cue responsive neurons (with more neurons responding to odor and tactile vs. sound and light stimuli) after cue-sucrose association does not depend on the number of sessions at final performance level.

*The authors could do a better job in the Results section of the text of explaining their criteria for a cue evoked response and for saying that a neuron did or did not fire to particular cues, as well as how associability was determined. For example, the examples in Figure 1 are very clear, but the criteria and rationale should be described clearly in the main text. Please spell out, as is done for some of the later analyses.*

We have revised the main text (see the second paragraph of the “Representation of cross- modal stimuli in GC” section of the Results) and added a more detailed description of the method used to determine neuron responsiveness to cues. We also added more details on how associability was determined.

*The authors note in the Discussion (subsection “Cross-modal integration in the gustatory cortex”) that the GC may be an ideal system to look for cross-modal coding. Leaving aside my question above about this, does this suggest that the findings here may not generalize to other primary association regions? Visual cortex? What about the olfactory system? I believe there are studies showing a variety of responses after training in olfactory bulb and piriform cortex to non-olfactory cues, as well as the visual cortex. This should be expanded on in the Discussion.*

We agree with the reviewer. We expanded the Discussion (see “Cross-modal integration in the gustatory cortex” section of the Discussion) to comment on the generality of our results and added references to the olfactory system and visual cortex.